# Intervention Effects of the Health Promotion Programme “Join the Healthy Boat” on Objectively Assessed Sedentary Time in Primary School Children in Germany

**DOI:** 10.3390/ijerph17239029

**Published:** 2020-12-03

**Authors:** Susanne Kobel, Jens Dreyhaupt, Olivia Wartha, Sarah Kettner, Belinda Hoffmann, Jürgen M. Steinacker

**Affiliations:** 1Division of Sports and Rehabilitation Medicine, Ulm University Hospital, 89075 Ulm, Germany; olivia.wartha@uni-ulm.de (O.W.); sarah.k.weber@gmx.de (S.K.); belinda.hoffmann@uni-ulm.de (B.H.); juergen.steinacker@uniklinik-ulm.de (J.M.S.); 2Institute for Epidemiology and Medical Biometry, Ulm University, 89075 Ulm, Germany; jens.dreyhaupt@uni-ulm.de

**Keywords:** inactivity, sedentary behaviour, sitting, school-based, childhood

## Abstract

Sedentary behaviour (SB) in children is related to negative health consequences that can track into adulthood. The programme “Join the Healthy Boat” promotes reduced screen time and a less sedentary lifestyle in schoolchildren. This study investigated the effects of the programme on children’s SB. For one year, teachers delivered the programme. A total of 231 children (7.0 ± 0.6 years) participated in the cluster-randomised study; there were 154 one year later at follow-up. Children’s SB was assessed using multi-sensor accelerometery, screen time via parental questionnaire. Effects were analysed using (linear) mixed effects regression models. At baseline, children spent 211 (±89) min daily in SB, at follow-up 259 (±109) min/day with no significant difference between the intervention (IG) and control group (CG). SB was higher during weekends (*p* < 0.01, for CG and IG). However, at follow-up, daily screen time decreased in IG (screen time of >1 h/day: baseline: 33.3% vs. 27.4%; follow-up: 41.2% vs. 27.5%, for CG and IG, respectively). This multi-dimensional, low-threshold intervention for one year does not seem to achieve a significant reduction in children’s SB, although screen time decreased in IG. Therefore, it should be considered that screen time cannot be the key contributor to SB and should not solely be used for changing children’s SB. However, if screen time is targeted, interventions should promote the replacement of screen time with active alternatives.

## 1. Introduction

There is ample evidence that regular and sufficient physical activity, as well as low levels of sedentary behaviour, are important for growth and health in school-aged children and youth [1,2,3], and contribute to the avoidance or limitation of risk of many conditions for chronic or degenerative diseases, which can occur even into adulthood [4,5]. Insufficient physical activity and too much sedentary time, on the other hand, have been associated with poor physical and mental health [6,7,8]. To investigate such associations, it is important to distinguish between sedentary behaviour and physical (in)activity. In adults, sedentary behaviour is associated with an increased risk of chronic conditions, including cardiovascular disease, type 2 diabetes and all-cause mortality [9,10,11,12]. In children, an increased risk of obesity, impaired glucose metabolism, cardiovascular disease, high blood pressure and cholesterol, depression and anxiety have been associated with sedentary behaviour [6,7,8,13,14]. Although evidence of such associations of sedentary time in children is mainly limited to screen time [6], sedentary behaviour has been shown to track into adolescence and adulthood [15,16].

Environmental changes and advances in technology, however, have resulted in sedentary behaviour being present within all age groups and settings of daily life. In some countries, children and adolescents spend a substantial amount of their time being sedentary, including screen time [17,18], which incorporates all screen-related activities such as television, DVD/video, smart phones, tablets and playing computer or video games [19]. Furthermore, the recent lockdown measures undertaken by several countries following the COVID-19 pandemic and the distance learning methods resulting from the closure of schools increased the sedentary and screen time among children and adolescents around the world [20].

Whilst, to date, there are no guidelines for overall sedentary time for children, there are recommended time limits for screen time based on prior evidence of poorer health and educational outcomes associated with this particular sedentary behaviour [2,14,21,22,23,24]. Whereas Canadian and Australian guidelines suggest limiting sedentary recreational screen time for children aged 5–17 years to no more than 2 h per day [25,26], the World Health Organization (WHO) thus far has only issued such guidelines for children up to 5 years of age [27]. In Germany, national guidelines for primary school children recommend spending as little time as possible with screen media, with a maximum amount of 60 min per day [28].

Research shows that children exceed those screen time recommendations [29,30,31,32,33,34], and studies investigating health outcomes related to screen time indicate that high screen time is associated with increased weight gain and obesity [30,35,36,37,38,39]. Consequently, reducing children’s sedentary time (including screen time) may be important for the prevention of chronic diseases even in adulthood [40]. Training interventions concerning the knowledge of the main risk factors for health and the adoption of healthy lifestyles conducted on school-age children, for instance, have previously been shown to be effective [41,42].

In order to do so, in South-West Germany, the setting-based health promotion programme “Join the Healthy Boat” promotes, amongst others, reduced screen media use and a less sedentary lifestyle in primary school children by incorporating activity breaks and offering more physical activity. The teacher-based programme supports and structures already present elements of the daily school routine such as educational lessons, physical activity sessions, and (family) homework. Against the background of the already identified need for early prevention, this research investigated the effectiveness of the programme by objectively assessing primary school children’s sedentary time longitudinally.

## 2. Materials and Methods

### 2.1. Intervention and Study Design

The school-based, teacher-centred health promotion programme “Join the Healthy Boat”, which was developed as a multi-dimensional intervention using Bartholomew’s Intervention Mapping Approach [43,44], is based on Bandura’s social cognitive theory [45] and Bronfenbrenner’s social ecological model [46]. The state-wide programme uses a train-the-trainer approach to disseminate and to enable primary school teachers to change the school environment as well as their teaching to promote physical activity, a healthy diet, and active leisure time without screen media on a weekly basis in their classrooms with no added lessons. The hereby given intervention materials, which were developed in collaboration with a team of experienced teachers are fully integrated into the school curriculum focusing on health promoting behaviour and environment change. The main focus of this intervention is to promote a healthy and active lifestyle by offering a physical activity friendly environment and action alternatives children can choose from in order to be less sedentary and more active. The teaching units, which were implemented by the classroom teacher on a weekly basis, for one school year include 20 lessons teaching health-relevant topics such as “why does my body need physical activity?”, increasing awareness, and offering ideas and alternatives for children’s leisure time without the use of screen media. Thirteen of those lessons (which can be held more than once) focus on physical activity promotion and a reduction of sedentary behaviour. Additionally, every day, two short activity exercises of five to ten minutes each were introduced into the children’s school routine in order to break up sedentary behaviour at school. To also involve parents into the programme, letters and so called family homework were handed out regularly, in order to get children to solve exercises actively together with their parents (further details: see [47]). Teachers recorded their implementation rates and only classes with at least 80% implementation were included in the sample.

To evaluate the programme in South-West Germany, a prospective, stratified, cluster-randomised, and longitudinal study with an intervention and control group was designed. Stratification of randomisation was carried out on grade level based on information about the distribution of participating teachers within the different schools. Stratification according to number of classes and grade levels was realised on six different levels. Cluster-randomisation was carried out on school level into intervention and control group. More details on the intervention’s design, theory-guided development and recruitment can be found elsewhere [44,48]. The University’s Ethics Committee approved the study (application no. 126/10), as did the Ministry of Culture and Education and it was conducted in accordance with the declaration of Helsinki. The study is also registered at the German Clinical Trials Register (DRKS-ID: DRKS00000494).

### 2.2. Participants

Participation in the programme was voluntary and participating teachers had to agree with the randomisation process. Children provided their assent, parents their written, informed consent to take part in the study as well as a separate consent for their children to wear a multi-sensor device assessing sedentary time objectively for six consecutive days.

Baseline data of 1947 primary school children (aged 5 to 8 years) taking part in the evaluation of “Join the Healthy Boat” were available; for the collection of objectively assessed sedentary time, a sub-sample of 231 children (7.0 ± 0.6 years, 46% male, intervention *n* = 133, control *n* = 98, 12% of the whole cohort) was investigated, including those children who agreed to wear a multi-sensor device and showed valid data of at least three days of more than 10 h of recorded data per day. A total of 154 children provided valid data of their sedentary time one year later at follow-up (8.0 ± 0.6 years, 46.1% male, intervention *n* = 102, control *n* = 52, 67% of the baseline sub-sample). Neither the sub-sample who agreed to objective measurements, nor the follow-up sample (sub-sample minus drop-out) differed from the whole sample with regards to age, gender and other socio-economic background data.

After baseline measurements had been taken, the intervention started in the intervention group only, the control group followed the regular school curriculum. Follow-up measurements were taken after one school year, directly after a six-week summer break with no intervention.

### 2.3. Measurements

During a school visit, anthropometric measurements (children’s height and body mass) were taken by trained staff to ISAK standards [49] using calibrated electronic scales and a stadiometer (Seca 862 and Seca 213, respectively, Seca Weighing and Measuring Systems, Hamburg, Germany). Children’s BMI was calculated and converted to BMI percentiles using national reference values [50]. Subsequently, children were classified into under-/normal weight (percentiles <90), overweight (percentiles ≥ 90) and obese (percentiles ≥ 97).

Sedentary time was assessed using a multi-sensor device (Actiheart^®^, CamNtech, Cambridge, UK), which was validated for children [51]. The chest-fitted sensor was worn for 24 h a day on six consecutive days. Recording intervals were set to 15 secs and to be included in the sample, at least 10 h per day including at least two weekdays and one weekend day had to be available [52]. First and last day of recording were excluded from the analysis (in order to antagonise a novelty factor on the first day, the last day never showed 10 h of recording). Heart rate and one-dimensional bodily acceleration were recorded. Energy expenditure (MET) was calculated using Actiheart^®^’s captive software [53]. Sedentary time was defined as “any waking behaviour characterised by energy expenditure ≤1.5 metabolic equivalents (METs)” [54]. Individual sleeping time was determined for every analysed day by an obvious increase and drop of heart rate for the point of awakening and falling asleep, respectively. This was subsequently subtracted from daily assessed recording time to quantify children’s sedentary time during waking hours only. In order to calculate total sedentary time, the available days were extrapolated to a full week, using a ratio of 5:2 for weekdays and weekend days.

Children’s screen time (including time spent with television and computer/game console) for weekdays and weekend days as well as their socio-demographic information were collected via a parental questionnaire. The included questions were based on the German Health Interview and Examination Survey for Children and Adolescents, which assessed health behaviour in 18,000 German children and adolescents [55]. Screen time was assessed as an ordinal variable with seven categories and dichotomised according to national guidelines [28] into one hour or less and more than one hour. Parental education level was determined based on the highest school education of either one parent or the single parent and thereafter dichotomised into tertiary education level (high school) and primary/secondary education level. Children were classed as having a migration background if at least one parent was born abroad or the child was spoken to in a foreign language during the first three years of life.

### 2.4. Data Analyses

Linear mixed-effects regression models and mixed-effects regression models for a binary outcomes were used to analyse group differences and in the examination of differences between weekdays and weekend days. Gender differences were also examined using mixed-effects regression models adjusting for age and BMI percentiles. Intervention effects were analysed using linear mixed effects regression models as well as mixed effects regression models for binary outcomes, controlling for age, gender, BMI percentiles, migration background, family education level, and baseline values. Descriptive statistics for continuous variables were displayed in mean values and standard deviations. Categorical variables were described with absolute and relative frequencies. To detect differences between the groups and samples, the two-sample t-test was used for continuous data, the chi^2^ test or Fisher’s exact test was used for binary data. Statistical analyses were performed using SPSS Statistics 25 (SPSS Inc., Chicago, IL, US) and SAS, version 9.4 (SAS Institute, Cary, NC, US) with a two-sided significance level set to α ≤ 0.05.

## 3. Results

A summary of participant’s descriptive data can be found in Table 1. There was no difference regarding age, body weight and height, BMI percentiles, migration background, and parental education level between the here analysed sub-sample, who agreed to objective activity assessment and total sample, nor between control and intervention group (descriptive data of the whole sample were published elsewhere [56]).

### 3.1. Sedentary Time

At baseline, children spent 211 (±89) min daily in sedentary time; at follow-up this increased to 259 (± 109) min per day, with no significant difference between intervention and control group. Sedentary time was significantly higher during weekends, compared to weekdays (*p* < 0.01, for control and intervention as well as baseline and follow-up; see Table 2).

At follow-up, no gender difference was observed for total sedentary time, nor was there a gender difference for sedentary time at weekdays and weekends, separately. Similarly, children’s weight status was not associated with differences in objectively assessed sedentary time. Children of parents with tertiary-level education (compared to primary- and secondary-level education), however, showed significantly more sedentary time on weekdays (285 ± 112 min/day vs. 225 ± 96 min/day, *p* = 0.01), but not on weekends or during the total week. On the other hand, having a migration background was significantly associated with more sedentary time at the weekend (346 ± 138 min/day vs. 276 ± 110 min/day, *p* < 0.01), but not at weekdays or during the total week.

Further, the final mixed effects regression model controlling for baseline values showed no significant intervention effects on total sedentary time, sedentary time at weekdays and on weekends for the total sample, nor for screen time (Table 3). When controlling for baseline values, gender, migration background as well as weight status and family education level, different results were only found for daily screen time during the whole week (*p* = 0.02).

### 3.2. Daily Screen Time

At baseline, 30.0% (*n* = 200, missing = 31) children spent more than one hour daily with screen media, with a significant difference between control and intervention group for daily screen time. At follow-up this amount increased to 32.4% (*n* = 46; see Table 2), with a significant difference between intervention and control group for daily screen time. Daily screen time was significantly higher during weekends, compared to weekdays (*p* < 0.01, for control and intervention as well as baseline and follow-up).

At follow-up, when assessed via mixed model analyses, parent reported daily screen time decreased in the intervention group, compared to the control group (daily screen media use of <1 h: 41.2% vs. 27.5%, for control and intervention group, respectively). This is especially visible (even though not significant) on weekends (62.8% vs. 49.5%, for control and intervention group, respectively). 

Assessing change for one year (as shown in Figure 1), most of the children in control and intervention group did not change their screen time behaviour. A total of 73.3% of children in the control group and 77.6% of children in the intervention group showed the same amount of screen time at baseline and at follow-up. Additionally, 10.6% of children in the intervention group decreased their daily screen time, whereas 8.9% of children in the control group used less screen media at follow-up. Respectively, 11.8% of children in the intervention group increased their screen time at follow-up compared to 17.8% in the control group.

Further, there was a significant difference in children using screen media for more than one hour daily when comparing weekdays with weekends (F = 9.70, *p* < 0.01), as well as for gender with boys reporting significantly higher screen times than girls (F = 5.90, *p* = 0.02). Moreover, screen times of children with migration background differed significantly from those without migration background. Children with migration background showed significantly more often screen times of more than one hour per day compared to children without migration background (F = 4.03, *p* = 0.05).

## 4. Discussion

This study aimed to evaluate the effectiveness of the setting-based health-promotion programme “Join the Healthy Boat” on sedentary time in primary school children. After one year, objectively assessed sedentary behaviour of children in the intervention group did not differ from those in the control group. In line with previous research on multi-component school-based interventions [57], sedentary time increased in the space of 12 months; this was especially evident at weekends. This increase is most likely due to the age related incline that is seen in sedentary time during childhood [58] and participation in “Join the Healthy Boat” did not manage to prevent this increase.

The programme intended to tackle sedentary behaviour at school with daily exercise breaks and the offer of action alternatives for children’s leisure time so they would refrain from using screen media after school or at the weekends. In order to involve their parents, family homework was issued regularly to encourage “screen free” days or weekends, offered ideas to where to go at the weekend or what games to play to get more physically active and break up sedentary time.

It has previously been suggested that sedentary time can be a predictor of chronic disease independent of physical activity levels [59]. Especially crucial are increased total sedentary time but also prolonged uninterrupted blocks of sedentary time. Both have been addressed in “Join the Healthy Boat” since the introduction of short activity exercises of five to ten minutes each was supposed to break up sedentary time at school, which should have happened in the middle of lessons, so the children were likely to be sitting before and after their exercise breaks. Despite this, this study shows an increase of 48 (± 20) min in average daily sedentary time, independent of control or intervention group.

A Scottish study has also tried to implement an additional 15 min of exercise into primary school children’s daily classroom routine. In “The Daily Mile” [60], teachers were asked to send the children outside (at a time of the teacher’s choosing) so they could exercise for 15 min at a self-selected pace and break up sedentary time. That intervention showed a significant reduction of sedentary time of around 18 min after half a year, with authors suggesting that children replaced sedentary time with physical activity [60].

More comparable results to the ones reported here were found in a one-year school-based intervention study in the Netherlands [57] as well as a recent Scandinavian study in first graders, intervening on teacher level in an after school care setting [61]. Similar to this study, teachers in Norway were trained to create a physical activity supporting environment in order to promote physical activity and reduce sedentary time. After seven months of intervention, no significant intervention effects were found on sedentary time. Moreover, from baseline to follow-up, total sedentary time in the intervention group increased slightly compared to the control group [54], which was observed in this study as well. Possibly a more intense intervention would have shown more effective results, such as seen when replacing standard desks with sit–stand desks [38]. Teachers encouraged children to use those desks for more than one hour per day, which reduced children’s weekday sitting time by half an hour after 4.5 months [40].

Here, children accumulated less sedentary time on weekdays compared to weekends, which goes in line with recent results of a large Finnish sample of secondary school children [62]. This was especially true for children with migration background, which has been shown previously [63]. It was argued that children, but more so their parents, are hardly accessible and, therefore, intervening on child and parental level is more difficult. It has also been suggested, that there are significant associations between parental activities and those of their children, including sedentary behaviour [64,65], with mothers’ and fathers’ time spent sedentary positively correlating with children’s sedentary time [65,66]. Parental influence on children’s health behaviours is a well-known determinant of childhood obesity [67] and a recent review has once more highlighted that parental involvement in addressing health issues such as sedentary behaviour in children is essential [68]. According to the authors, face-to-face meetings with parents appeared to show greater effects compared to handed out written material [68]. In “Join the Healthy Boat”, parental involvement was realised by five parent letters in three different languages throughout the year (three of which addressed physical activity and sedentary behaviour), six family homework assignments, and two parents’ nights at school. Possibly, a more active and more frequent involvement of parents in joint activities at school or regular face-to-face contact between teachers and parents would have shown more positive effects.

However, since sedentary behaviour was addressed by increasing physical activity (offers) and reducing screen time (in class and with the parents), this might have led to the observed decrease in children spending more than one hour per day with screen media on weekdays. Compared to other European cohorts, Germany shows generally low screen time values [69]; yet, this sample showed very low screen time, even initially. This national cut-off of one hour per day [28] to determine screen time showed that most children adhered to those guidelines, which is unusually high, even when compared with studies using a cut-off of two hours per day [69]. Screen time was assessed via parental questionnaire (as in other large European cohorts [69]) and, therefore, subject to social desirability and recall bias. Nonetheless, the difference in parental-reported screen time of children in the control and intervention group indicates a positive tendency of the programme.

It is well known that screen time is associated to adiposity, quality of life, mental health, and sleep outcomes [35,70,71]. Yet it has been shown that high sedentary time is not necessarily due to increased screen time [72,73]. Comparing children with low, medium, and high sedentary time showed no difference in daily screen time but when comparing it to their total sedentary time, percentages ranged between 27% and 71% of total sedentary time [73]. Therefore, it should not solely be intervened to reduce children’s screen time in order to decrease their sedentary time.

Still, primary schools have previously been described as an ideal setting for interventions to reduce sedentary behaviour through practice, policy, and a supportive environment [68,74]; yet, this requires an active involvement not just of parents but also of teachers. This sometimes proves difficult, especially if the teacher’s own behaviours do not agree with the messages delivered in the intervention [75]. It has been suggested that school-based interventions are most effective if children have relevant role models—from parents and teachers [68].

One of the challenging aspects of this intervention was the transmission of the lessons’ content to children’s behaviour at home. As previously mentioned, family involvement was limited to thirteen times during one school year, with hardly any personal contact between teachers and parents. Although, implementation was recorded on teacher level, it is impossible to say whether the parents actually received or read the handed out written information. Further, possibly more individual recommendations on school level, such as school-specific plan for situational prevention worked out jointly with teachers, school head and the project team would have led to greater commitment and might have had greater impact on the structural changes schools were offered to make in order to get children more active and to reduce their sedentary time at school. Possibly a one-size-fits-all solution is not necessarily effective at tackling sedentary behaviour in a school setting.

In order to design future health promotion programmes that incorporate all those aspects and implement findings from this research, some limitations have to be considered when interpreting these results. Sedentary behaviour was (although assessed objectively) calculated on the basis of energy expenditure, which might have led to misinterpretations in some children. Screen time, on the other hand, was assessed via parental report and, therefore, as mentioned before, is subject to social desirability and recall bias. Further, due to the voluntary agreement of parents, teachers and children to participate in this study, a selection bias cannot be ruled out, which also led to the results not being representative although the sample was spread over a relatively large area. The sample also showed very low rates of screen time initially, which possibly should have been assessed before trying to intervene in that sample. Besides, it should be noted that follow-up took place after a six-week summer break with no intervention, which might have led to the lack of intervention effects. Additionally, possibly a longer lasting, more intense intervention with extra lessons and especially more alternatives for at home would have shown more positive effects. Furthermore, the lack of intervention effects on children’s sedentary time could also be due to the study design. The voluntary participation and the randomisation into intervention and control group might have made control schools susceptible for other health promotion, which may have caused changes in children’s sedentary behaviour in the control group. However, this design and its randomisation is also a strength of this study, so it the relatively large sample size and the intervention duration of one year. Additionally, the use of a solid theoretical framework as basis of this intervention should have helped to generate more positive intervention effects [76], as should the combination of implementing environmental changes, the offer of action alternatives and parental involvement [41,77].

## 5. Conclusions

This multi-dimensional, low-threshold intervention for one year does not seem to achieve a significant reduction in children’s objectively assessed sedentary time. A more intense or specific intervention providing increased opportunities to break up sedentary time at school and a better transfer to children’s homes might possibly have led to positive intervention effects and should be considered for the future. Additionally, sedentary time was higher at weekends, which calls for a greater parental involvement, which should be part of any well-planned health promotion programme. However, parent-reported screen time decreased in the intervention group, which leads to the assumption that screen time cannot be the key contributor to sedentary time and should not solely be used for changing children’s sedentary behaviour. Thus, if screen time is targeted, interventions should promote replacing screen time with alternatives in order to get children engaged in more physical activity. Hence, health behaviours need to be changed in more detail and with a greater intensity, especially at the family level, where the delivery of action alternatives and indirect offers for physical activity and the reduction of sedentary time may not take effect.

## Figures and Tables

**Figure 1 ijerph-17-09029-f001:**
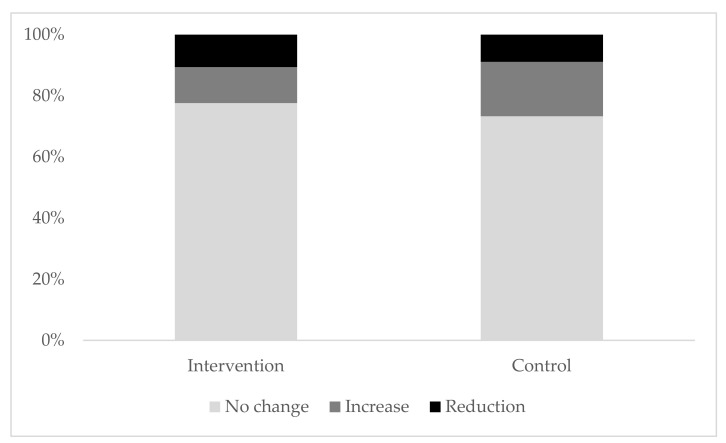
Change in screen time over one year in intervention and control group in percent.

**Table 1 ijerph-17-09029-t001:** Descriptive characteristics of total sample, control and intervention group at follow-up.

Variables	Missing Values	Total Sample (*n* = 154)	Control (*n* = 52)	Intervention (*n* = 102)
Gender (male); *n* (%)	0	71 (46.1)	22 (42.3)	49 (48.0)
Age (years); m ± SD	0	8.0 ± 0.6	8.0 ± 0.6	8.0 ± 0.7
Height (cm); m ± SD	0	129.5 ± 6.5	129.9 ± 6.5	129.2 ± 6.5
Weight (kg); m ± SD	1	27.0 ± 4.9	26.7 ± 3.5	27.2 ± 5.4
BMIPCT m ± SD	1	45.5 ± 26.6	44.0 ± 23.4	46.3 ± 28.2
Weight status				
Overweight/Obese; *n* (%)	1	10 (6.5)	1 (1.9)	9 (8.8)
Migration background; *n* (%)	20	37 (27.6)	9 (19.6)	28 (31.8)
High family education level; *n* (%)	25	45 (34.9)	14 (33.3)	31 (35.6)

Values are mean (m) ± standard deviation (SD) or numbers (*n*) and percentages (%).

**Table 2 ijerph-17-09029-t002:** Time at baseline and follow-up (in rounded min) spent in sedentary time (without sleep) and percentage of children using screen media for more than one hour per day.

Variables	Missing Values	Total Sample	Control	Intervention
		*n* = 231 (b), 154 (f)	*n* = 98 (b), 52 (f)	*n* = 133 (b), 102 (f)
Sedentary Time (min/day) (mean ± SD (95% CI))				
Total week				
Baseline	0	211 ± 89 (199–222)	219 ± 87 (201–236)	205 ± 91 (189–220)
Follow-up	0	259 ± 109 (242–276)	254 ± 99 (226–281)	262 ± 115 (239–284)
Weekdays				
Baseline	0	199 ± 95 (187–212)	207 ± 92 (188–225)	194 ± 98 (177–211)
Follow-up	2	241 ± 108 (224–258)	240 ± 102 (211–268)	242 ± 112 (219–264)
Weekend				
Baseline	0	239 ± 103 (226–253)	249 ± 102 (228–269)	233 ± 104 (215–250)
Follow-up	1	295 ± 123 (275–315)	293 ± 125 (258–328)	296 ± 124 (271–320)
Screen Time of >1 h/day (*n*(%) (95% CI))				
Total week				
Baseline	31	60 (30.0) (23.7–36.9)	29 (33.3) (23.6–44.3)	31 (27.4) (19.5–36.6)
Follow-up	12	46 (32.4) (24.8–40.8)	21 (41.2) (27.6–55.8)	25 (27.5) (18.6–37.8)
Weekdays				
Baseline	31	15 (7.5) (4.3–12.1)	11 (12.6) (6.5–21.5)	4 (3.5) (1.0–8.8)
Follow-up	12	10 (7.0) (3.4–12.6)	2 (3.9) (0.5–13.5)	8 (8.8) (3.9–16.6)
Weekend				
Baseline	30	93 (46.3) (39.2–53.4)	39 (44.8) (34.2–55.9)	54 (47.4) (37.9–56.9)
Follow-up	12	77 (54.2) (45.7–62.6)	32 (62.8) (48.1–75.9)	45 (49.5) (38.8–60.1)

Values are mean (m) ± standard deviation (SD) and 95% confidence interval (CI) or numbers (*n*) and percentages (%) and 95% confidence interval (CI). (b) Baseline; (f) follow-up.

**Table 3 ijerph-17-09029-t003:** Final mixed effects regression model for binary outcomes.

Variables	Total Sample (*n* = 154)		
	*n*	Estimate (CI 95%)	*p*
Sedentary Time incl. baseline and intervention/control group			
Total week	100	−35.6 (−91.1;20.0)	0.21
Weekdays	99	−23.4 (−87.6;40.8)	0.47
Weekend	99	−22.0 (−87.9;43.9)	0.51
Screen Time of >1 h/day incl. baseline and intervention/control group			
Total week	130	0.13 (−0.02;0.29)	0.08
Weekdays	130	−0.04 (−0.13;0.05)	0.42
Weekend	130	0.08 (−0.09;0.24)	0.36

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
