# Peer review of "Intervention Effects of the Health Promotion Programme “Join the Healthy Boat” on Objectively Assessed Sedentary Time in Primary School Children in Germany"

_ijerph, 2020, doi:10.3390/ijerph17239029_

Round 1

Reviewer 1 Report

This manuscript submitted by Kobel et al, attempted to elucidate effect of health promotion programme on children's sedentary time. The present study surely will provide some interesting information on the social level that will help parents to prevent the onset of obesity or over weight for their children's, below are some minor concerns I have for this manuscript. 

Minor Comments-

  1. In the result section P5 (L173-L175), author mentioned the migration background is significantly associated with sedentary time? Here author should provide the reason and discuss this impact of background migration.
  2. Author should also mention what are the gadgets that are included in the screen time. Please mention in material and method section.

Reviewer 2 Report

Thank you for giving me the opportunity to review your paper. The content of the paper is of great interest for all people dealing with promoting less sedentary behaviour (SB) and more physical activity for youth. Although the intervention did not successfully reduce SB, it can help to understand the complexity of SB and to bring up successful interventions.

In my opinion, a few aspects have to be addressed before publication of the paper.

General remarks:

  • As you conducted a longitudinal study, please also provide baseline values for screen time, sedentary time etc. in tables and/or text. In my opinion, this is the most important and major concern.
  • When discussing the effectiveness of interventions running in many schools, a key challenge is the implementation of the intervention. I strongly encourage the authors to provide information about quantity and quality of the implementation in the paper, at the moment there is no information included.
  • Please avoid multi-referencing of single arguments, in order to reduce the number of references
  • “;” are used extensively, I assume to shorten some sentences. Please carefully check if “;” are used according to grammar rules.
  • The paper mostly used mean ± SD for the results, I would like to encourage using mean and 95%CI, instead

Abstract:

  • Line 16: Please make clear that the follow-up measures were conducted after 12 months of intervention. Or was there a gap between the 12 months intervention and the FU?
  • Line 21: In the results section, please also add baseline values
  • Lines 24-25: I agree that it is not all about screen time when we aim to reduce sedentary time. However, I think we should provide some options for being active/not sedentary when we reduce screen time. Therefore, I would suggest to add something like “…key contributor to ST, and if screen time is aimed for, interventions should promote to replace screen time with activity options.”

Introduction:

  • Line 61: As join the healthy boat seems to be a complex interventions, it would be useful to know, what proportion of the intervention is dedicated to reducing screen time and a less sedentary lifestyle, compared to the other components of the interventions. This could also be addressed in 2.1, Intervention design

Methods

  • Line 112: Please clarify how you did assess that there were no differences, because these data is not given in the upcoming tables. Secondly, did you compare the subsample of this study to the full study, or did you compare the n=154 of the FU to the n=231 at baseline of the subsample?
  • Lines 136f: please provide information/references on the questionnaires you utilized

Results

  • Tables in general: at the moment, some of your written results (and discussions later) are not supported by the data included in your results. In my point of view, lots of the potential of the study is lost due to data not shown.
  • Table 1: please add descriptive data of the full study cohort (n=1947), there is enough space for an additional column. 7% and 12% difference in overweight/obese and migration background are no difference?
  • Table 2: as you conducted a longitudinal study, I would expect baseline values for sedentary time and screen time in table 2. Again, there is enough space for an additional column.
  • Lines 182f: Please provide the absolute amount of screen time for both groups. You may also want to add another table.
  • Lines 188f: In my opinion, transitions in groups over time are better provided in a figure than in written text. I attached a sample figure from Jago et al. 2018.
  • I would like to get more information about the implementation/process evaluation. For example: how many lessons were dedicated to sedentary time/screen time? Did the lessons follow the intervention protocol? What proportion of the children received which proportion of the lessons? If there is no information available, this has to be addressed in the limitations of the study.

Discussion

  • The increase/decrease of screen time/sedentary time on weekdays/weekends is not supported because this data is not (yet) shown.
  • The extensive discussion of other studies could be reduced. Instead, you could expand on the aspect on how to transmit the content of school lessons to sedentary time at home, the involvement of parents, and a discussion of the lessons learned of your study, with consequences for improvement/future interventions

Reviewer 3 Report

The authors present results of a school-based intervention program attempting to foster increased physical activity in school age children, and conclude that children in the program (compared to control children not in the program) had no change in their sedentary time or screen time. My main concern is that there is a relatively small subset of children for whom the "sedentary time" data was collected, very likely not a large enough sample size for this type of "low threshold" intervention. The authors do not relay their sample size calculation, or what they determined would be a significant change in sedentary activity. Moreover, as the authors themselves reported in the discussion, at baseline these children had very low levels of screen time usage and sedentary time (approx 4.5 hours/day which is less than a typical school day instruction, during which time I assume children are sedentary). The population also had almost no overweight/obesity---focusing on a population that is already healthy, and exhibiting the desired behaviors, does not make sense for an intervention study.

The presentation of the study and the data was difficult to follow. "ST" is a confusing shorthand as it can stand for "screen time" and "sedentary time". The data is offered only in tables---some graphical representations may be helpful. 

Reviewer 4 Report

The manuscript “Intervention Effects of the Health Promotion Programme "Join the Healthy Boat" on Objectively Assessed Sedentary Time in Primary School Children in Germany” provides the results of a study aimed to evaluate the effectiveness of the setting-based health-promotion programme “Join the Healthy Boat” on sedentary time in primary school-children. The study is very interesting and the manuscript is adequately organized and written. In my opinion, only a few minor corrections should be made.

Introduction

Sufficient physical activity as well as low sedentary lifestyle not only contribute to a healthy growth of children but also to reduce the risk conditions for developing chronic diseases even in adulthood. Therefore in line 30 please add the following sentence supported by the relevant literature:: “…and contribute to avoid or limit many risk conditions for chronic or degenerative diseases which can occour even in adulthood” [Panico et al. IJERPH 2020,17:1208; Dietz et al. Chronic Disease Prevention: Tobacco Avoidance, Physical Activity, and Nutrition for a Healthy Start. JAMA.2016;316:1645–1646.]

Lines 33-36: The following sentence “here, sedentary behaviour is defined as “any waking behaviour characterised by energy expenditure ≤1.5 metabolic equivalents (METs) while in a sitting or reclining posture” [7]” should be deleted as the definition of sedentary time is explained, as it should be, in the materials and methods section.

Line 46: In consideration of the recent lockdown periods undertaken in several countries following the Covid-19 pandemic and the possibility that children and adolescents have increased their sedentary and screen times, you could add the following sentence and report the supporting literature: “Furthermore, the recent lockdown measures undertaken by several countries following the Covid-19 pandemic and the distance learning methods resulting from the closure of schools increased the sedentary and screen time among children and adolescents around the world” [Rundle et al. "COVID‐19–Related School Closings and Risk of Weight Gain Among Children." Obesity (2020)].

Line 59: I would suggest adding the following sentence: "The training interventions concerning the knowledge of the main risk factors for health and the adoption of healthy lifestyles conducted on school-age children were often effective” [Lloyd et al. "Effectiveness of the Healthy Lifestyles Programme (HeLP) to prevent obesity in UK primary-school children: a cluster randomised controlled trial." The Lancet Child & Adolescent Health 2018,2:35-45; Carducci et al. Improving awareness of health hazards associated with air pollution in primary school children: Design and test of didactic tools. Applied Environmental Education & Communication 2016,15:247-260].

Materials and methods

Line 130: Please add what "MET" mean.

Line 149: “were” displayed

Line 150: “were” described

Results

The sentence at lines 168-169 is unclear; please rewrite.

Round 2

Reviewer 2 Report

I would like to thank the authors for responding to my comments. In my opinion, the majority of comments has been sufficiently addressed. Only three questions are left:

  • Very important: it seems that the numbering of the references is incorrect! For example, ref 56 (line 178) should be ref 65. I strongly encourage the authors to cross-check all references!!!
  • Table 2: thank you for adding baseline values. Comparing baseline to follow-up data would be best possible if you would provide baseline and follow-up data next to each other (currently, follow-up is presented four line below baseline data). Therefore, I would encourage the authors to check if another layout is more suitable for table 2.
  • Abstract: I am sorry that valid comments fail in their implementation due to restrictions in word counts. Please consider shortening other parts of the abstract, for example in the background section, to create space for results/discussion. Please mind that if someone else reviews papers, the title and abstract screening is prior to full-text reading. Therefore, the abstract should include all aspects that you consider valid/important.
